# Learning 3D Garment Animation from Trajectories of A Piece of Cloth

**Yidi Shao**[1]    **Chen Change Loy**[1✉]    **Bo Dai**[2,3]
[1]S-Lab, Nanyang Technological University
[2]The University of Hong Kong, [3]Shanghai Artificial Intelligence Laboratory
yidi001@e.ntu.edu.sg, ccloy@ntu.edu.sg, bdai@hku.hk

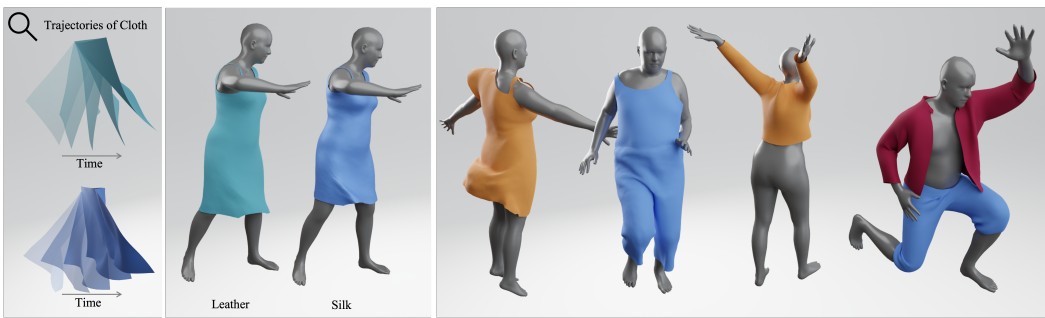

Figure 1: Given the observed piece of cloth as shown on the left, we aim to animate various garments inheriting the attributes from the observations as shown in the middle and right. We disentangle the garment-wise learning into two sub-tasks: 1) learning constitutive relations by our proposed EUNet; 2) animating diverse garments through energy optimization constrained by EUNet.

## Abstract

Garment animation is ubiquitous in various applications, such as virtual reality, gaming, and film production. Learning-based approaches obtain compelling performance in animating diverse garments under versatile scenarios. Nevertheless, to mimic the deformations of the observed garments, data-driven methods often require large-scale garment data, which are both resource-expensive and time-consuming. In addition, forcing models to match the dynamics of observed garment animation may hinder the potential to generalize to unseen cases. In this paper, instead of using garment-wise supervised learning we adopt a disentangled scheme to learn how to animate observed garments: 1) learning constitutive behaviors from the observed cloth; 2) dynamically animate various garments constrained by the learned constitutive laws. Specifically, we propose an Energy Unit network (EUNet) to model the constitutive relations in the form of energy. Without the priors from analytical physics models and differentiable simulation engines, EUNet is able to directly capture the constitutive behaviors from the observed piece of cloth and uniformly describes the change of energy caused by deformations, such as stretching and bending. We further apply the pre-trained EUNet to animate various garments based on energy optimizations. The disentangled scheme alleviates the need for garment data and enables us to utilize the dynamics of a piece of cloth for animating garments. Experiments show that EUNet effectively delivers the energy gradients due to the deformations. Models constrained by EUNet achieve more stable and physically plausible performance compared

---

✉Corresponding Author.

38th Conference on Neural Information Processing Systems (NeurIPS 2024).

with those trained in a garment-wise supervised manner. Code is available at
https://github.com/ftbabi/EUNet_NeurIPS2024.git .

## 1 Introduction

To create a realistic virtual world, it is crucial to ensure the clothes worn by digital humans have natural and faithful deformations similar to those in real life. Given its significance, garment animation has been extensively explored and applied in various fields, including filmmaking, virtual try-on, and gaming.

Recent deep-learning based methods [35, 2, 27, 22, 28] exhibit great potential in mimicking the dynamic patterns of the observed garments with compelling efficiency, generality and robustness. A common strategy of deep-learning based methods is to model garments as functions of SMPL-based human parametric models [35, 23, 2, 31, 4], which is inevitably limited to clothes of similar topologies of human bodies, such as T-shirt, failing to capture the dynamics of loose garments like dresses. To overcome such drawbacks of SMPL-based methods, there are attempts that adapt garment-specific designs [21, 20], and particle-based designs [28]. Nevertheless, to obtain similar deformation patterns as the observed garments, most existing deep-learning based methods require large-scale garment-wise datasets for training, which usually contains hundreds of samples, covering humans of various shapes and actions with garments of diverse topologies and materials. Collecting such a sophisticated dataset is both resource-expensive and laborious, especially when it is designed to ensure the generality and robustness of learned models, which may take months or even years to build. On the other hand, overly relying on a large-scale dataset may not be an efficient and effective way to obtain a good garment animation model that can generalize well to unseen garment topologies, human shapes and actions, as well as environmental factors.

To avoid the overhead and drawbacks of directly learning from large-scale garment-level datasets, in this paper we adopt a disentangled framework that learns to animate garments from *just a single piece of cloth*. A key observation is that while garments may vary in topologies, and their dynamics seem to be different given external forces, for garments and cloth made of the same material the constitutive relations, such as the stress-strain behaviors of elastic materials, remain consistent and are agnostic to garment topologies. Moreover, the dynamics of garments can also be defined as a time evolution problem constrained by both external forces and constitutive relations, which again is agnostic to garment topologies. Therefore, the task of mimicking garments' realistic dynamics is disentangled into two sub-tasks: 1) capturing the constitutive behaviors of some materials from a piece of cloth; 2) applying the learned constitutive laws to optimization-based scheme [9, 10] to dynamically animate garments made of the same material.

Learning constitutive laws is a non-trivial task. Domain experts design various analytical physics models to describe physical properties of certain materials, such as Piola Kirchhoff stress, bending model [6] by Bridson, and elastic models by St. Venant-Kirchhoff and neo-Hookean. Some methods [5, 34, 25, 11] take efforts to estimate the corresponding physics parameters, such as the Lame constant in St. Venant-Kirchhoff elastic model, to fit the analytical models to observed materials, while others [33, 37, 30, 32, 13, 18, 15] tend to rely on these analytical physics models as priors and embed them in the formulations. However, there is no general standard to choose the most suitable physics model automatically,and existing models cannot cover all materials. As a result, methods relying on these physics models are limited in generality, especially when handling garments whose physical properties are unknown and difficult to measure.

In this paper, we propose Energy Unit Network (EUNet), a method that learns to describe the constitutive laws in the form of energy directly from the observed trajectories. Our EUNet is able to uniformly generate the energy gradients caused by different deformations from a cloth system with dissipation, such as the deformations of stretching and bending and the damping effects due to the air. Specifically, to achieve the topology-independent modeling of constitutive relations, EUNet predicts edge-wise energy units, of which the summations represent the garment-wise energy. Each energy unit consists of potential and dissipation energy units, which are predicted by two sub-networks in EUNet respectively. While the observed sequences are unable to provide explicit supervision of constitutive behaviors, we propose to estimate the systems' total energy, including the dissipation energy, between adjacent frames to extract the variations of cloth's energy. With the system-wise scalars as supervising signals, we train EUNet directly from the observed trajectories without the

need of analytical clothing models or differentiable simulators as prior. Since different deformations could lead to similar energy in total, the system-wise scalars may confuse EUNet to obtain plausible energy gradients. To further reduce the ambiguity and improve the accuracy of our EUNet, we build vertex-wise contrastive loss, which takes inspiration from the energy optimization [9] that the evolution of the dynamics follows the path minimizing the system's total energy. Thus, we constrain the EUNet that the sampled vertex-wise disturbance, such as those caused by random noise, will lead to a larger energy in total. Once we obtain the pre-trained EUNet, we embed the predicted energy as a part of optimization-based physics loss [26, 10] to rollout the temporal evolution of garments.

The disentangled scheme and EUNet benefit garment animations in several aspects. First, the disentanglement enables us to use only a piece of cloth as training data to animate various garments, which is data-efficient compared with the large scale of garments dataset [2, 28]. Second, without auxiliaries of existing physics models as prior, such as St. Venant-Kirchhoff model that need extra estimations of the physics parameters to fit the target materials, EUNet is able to extract the constitutive laws directly from observed cloth and uniformly capture the potential energy derived from different deformations, such as stretching and bending. Third, since EUNet is agnostic to the topologies of clothes, EUNet can describe the deformation patterns of various garments, such as T-shirts and dresses, and naturally support the animation of different garments based on the energy optimization [26, 10]. Consequently, we bridge the gap between observing cloth and animating various garments inheriting the attributes of observed materials.

To train our EUNet, we collect the dynamics of a piece of square cloth made of common materials, i.e., silk, leather, cotton, and denim. The square cloth, whose corners on top are pinned, is initialized with random positions and deforms under the influence of gravity. To verify the quality of the animated garments constrained by our EUNet , we apply an optimization-based scheme [9, 26, 10] to solve the dynamic deformations. We compare with models trained in a garment-wise supervised manner on clothes of the same materials, and evaluate the Euclidean errors between the predictions and ground truth data. As shown in the experiment, models constrained by our EUNet delivers lower errors and physically plausible rollouts comparing with models trained directly in garment-wise manner, even for long-term predictions.

Our contributions can be summarized as follows: 1) we propose to learn the dynamics of observed garments through the disentangled scheme: learning constitutive relations and dynamically animating by energy optimizations; 2) we propose EUNet to directly capture the constitutive laws from observed trajectories using *just a single piece of cloth*; 3) we combine EUNet with optimization-based scheme to animate diverse types of garments inheriting the dynamic patterns from the observed cloth.

## 2 Related Work

**Learning-based Garment Animation.** Most existing methods model garments conditioned on the parameters of SMPL-based human bodies, including the human pose and shape [35, 2, 31, 4]. The over-reliance on the SMPL parameters inevitably leads to static models, which predict unique deformations for identical parameters of humans, garment-specific design [21, 23, 20], and limited generalization abilities to interact with obstacles beyond SMPL-based human models. While plausible deformations are obtained mostly on skinny garments, which have similar topologies as the underlying human bodies, previous work struggles in animating loose clothes. Extra designs are required to extend the human-dependent garment model to handle motions [27, 36] and loose garments [22]. Recently, by modeling the interactions among garments' vertices, simulation-based methods [10, 28] achieve topology-independent design and are highly generalizable to unseen garments and obstacles. However, most existing approaches adopt supervised learning and demand large-scale data of high quality. While some methods [3, 26, 10] avoid the need for data by using physics priors from analytical clothing models, the dynamic behaviors of predicted garments are restricted within the space defined by the clothing models and unable to mimic observed cloth with attributes beyond the known physics prior.

In contrast, we apply the disentangled scheme of the supervised task and utilize only a piece of cloth as ground truth data to learn the constitutive relations from observations. Constrained by the learned EUNet , we can directly animate garments of diverse topologies inheriting the attributes of observed cloth.

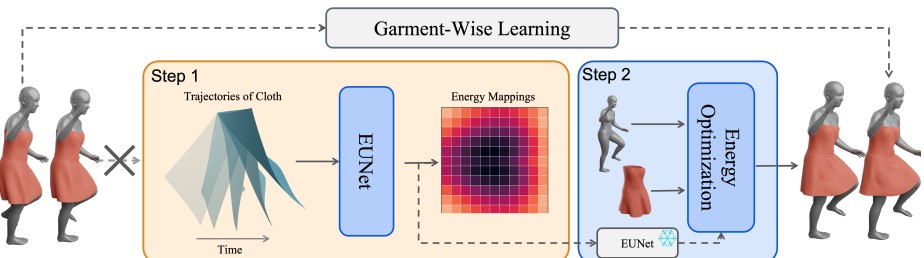

Figure 2: Overview of the disentangled learning scheme and our EUNet for garment animation. Unlike traditional garment-wise learning which relies on large scale of garment data, we first aim to capture the constitutive relations from the observed piece of cloth using our EUNet. Without the prior of analytical clothing models or differentiable simulator, EUNet is able to extract the potential energies of the cloth under different deformations, such as stretching and bending, directly from the observed trajectories in a system with dissipation. Secondly, given the external force sequences and the garment templates, we dynamically animate various garments based on the energy optimizations, where EUNet serves as material priors. As a result, we can animate garments that inherit the attributes, such as the stiffness, from the observed cloth, and achieve robust and physically plausible animations.

**Constitutive Laws.**   Constitutive laws describe the materials' reactions towards external stimuli, such as the potential energy or forces caused by deformations, and have been ubiquitous in the physics study: elasticity [1], plasticity [8], and fluid [7]. A common practice [5, 34, 25, 11] to mimic objects' constitutive behaviors is estimating physics parameters to fit the analytical physics models to the observed materials, such as predicting the lame constant in St. Venant-Kirchhoff elastic model. Other learning-based approaches [29, 33, 37, 30, 32, 13, 18, 15, 17] adopt analytical models as known physics priors, which are used to generate labeled data or embedded as part of the formulations. In addition, some methods [19] depend on differentiable particle-based physics engines to enable the training. In this paper, we focus on the elasticity behaviors of clothes represented by meshes. Unlike previous learning-based methods determined by analytical models or differentiable simulation engines, we formulate our constitutive relations constrained by general physics laws, such as the Lagrangian mechanism, enabling the training directly from observed data in a generic manner.

**Incremental Potential.**   Discretizing in time, the evolution of the objects' states, such as the positions and velocities, can be solved by minimizing the incremental potential of the system [12, 14, 16]. Recent methods [26, 10] follow the scheme and formulate a set of physics-based loss terms. Since minimizing the loss terms is equivalent to resolving the optimization problem, neural networks can serve as robust and differentiable tools to solve equations constrained by the incremental potential. In this paper, the constitutive laws learned by EUNet can be seamlessly embedded as part of the incremental potential, enabling to animate garments with attributes captured by our EUNet.

## 3   Methodology

The objective of mimicking the observed garment dynamics is to minimize the discrepancies between the predicted clothes and ground truth data under external forces, such as the motions of human bodies. In other words, the task seeks to capture the intrinsic deformation patterns of the observed garments, such as the stress-strain behaviors, which are independent of external influences. To achieve this, as shown in Figure 2, we disentangle the task into 1). learning the constitutive behaviors of several materials and 2). dynamically animating garments constrained by the learned laws through energy optimization. In Section 3.1, we introduce the required data for training, which consists of the trajectories of just pieces of cloth instead of the whole garment. Next, in Section 3.2, we direct the attention to our EUNet which captures the constitutive behaviors from the observed cloths. Most importantly, we propose a generic scheme to model the cloth's energy, without the need for an analytical clothing model or differentiable simulation engine. Finally, in Section 3.3, we describe the dynamic animations of garments constrained by our EUNet.

### 3.1   Data of A Piece of Cloth

Since constitutive laws are agnostic to the topologies, the dynamics observed in a piece of cloth are consistent with those exhibited by entire garments. Therefore, we select a square piece of cloth as the

subject of our observations, which is easier to design and more efficient to collect compared with garments. Specifically, we select clothes made of several materials typical in real life: silk, leather, denim, and cotton. We pin the cloth's two corners on top as shown in Figure 1. With randomly initialized positions and velocities, the cloth moves and deforms under the influence of gravity. The cloth dynamics are synthesized using Blender[0]. More details about the data can be found in the Appendix.

## 3.2 Capturing Constitutive Laws

**Problem Formulations.** We denote the mesh-based cloth with triangulated faces at time $t$ by $M^t = \{\boldsymbol{X}^t, \boldsymbol{E}\}$, where $\boldsymbol{X}^t = \{\boldsymbol{x}_i^t\}_{i=1}^N$ are the positions for $N$ vertices, and $\boldsymbol{E}$ is the un-directed edges of the mesh. The rest state of the cloth, which is the template, is denoted by $\bar{M} = \{\bar{\boldsymbol{X}}, \boldsymbol{E}\}$. The cloth's attributes, such as the stiffness, are represented by $\boldsymbol{a}$. Given observed trajectories of cloth $\{M^t\}_{t=0}^T$, we aim to predict the cloth's energy at time $t$ by

$$\Phi(\boldsymbol{X}^t, \boldsymbol{X}^{t-1}, \bar{\boldsymbol{X}}, \boldsymbol{E}, \boldsymbol{a}) = \Phi_p(\boldsymbol{X}^t, \bar{\boldsymbol{X}}, \boldsymbol{E}, \boldsymbol{a}) + \Phi_d(\boldsymbol{X}^t - \boldsymbol{X}^{t-1}, \boldsymbol{E}, \boldsymbol{a}). \tag{1}$$

$\Phi_p(\cdot)$ is the potential energy describing the constitutive behaviors of the cloth, whereas $\Phi_d(\cdot)$ is the dissipation energy caused by the damping effects in the dynamic process. For simplification, we represent the potential energy at time $t$ by $\Phi_p^t$ and the corresponding dissipation energy $\Phi_d^t$ in the following sections.

**Supervised Learning of Constitutive Behaviors.** Instead of modeling the constitutive relations conditioned on the entire cloth, we decompose the mesh-level energy $\Phi(\cdot)$ into edge-wise energy units $\phi(\cdot)$, which are modeled by our EUNet as follows:

$$\Phi_p^t = \sum_{e_{ij} \in \boldsymbol{E}} \phi_p\left(\Delta l_{e_{ij}}^t, \Delta \boldsymbol{\theta}_{e_{ij}}^t, \boldsymbol{a}\right), \tag{2}$$

$$\Phi_d^t = \sum_{e_{ij} \in \boldsymbol{E}} \gamma_{ij}^t \phi_d\left(\left\|\frac{\boldsymbol{x}_i^t + \boldsymbol{x}_j^t}{2h} - \frac{\boldsymbol{x}_i^{t-1} + \boldsymbol{x}_j^{t-1}}{2h}\right\|_2, \boldsymbol{a}\right), \tag{3}$$

where $\Delta l_{e_{ij}}^t = l_{e_{ij}}^t - \bar{l}_{e_{ij}}$ are the length differences between deformed edge $l_{e_{ij}}^t$ at time $t$ and template edge $\bar{l}_{e_{ij}}$. $\Delta \boldsymbol{\theta}_{e_{ij}}^t = [\alpha_{e_{ij}}^t, \beta_{e_{ij}}^t] - [\bar{\alpha}_{e_{ij}}, \bar{\beta}_{e_{ij}}]$, where $[\alpha_{e_{ij}}^t, \beta_{e_{ij}}^t]$ and $[\bar{\alpha}_{e_{ij}}, \bar{\beta}_{e_{ij}}]$ represent the relative angles between $i$-th and $j$-th vertex normals in deformed states and rest states respectively. In practice, we represent all angles $\theta$ by $[\cos\theta, \sin\theta]$ for convenience. $\gamma_{ij}^t$ is the summation of the areas of the faces with a common edge $e_{ij}$. $h$ is the time interval between adjacent frames. Please refer to the Appendix for more details.

The decomposition into energy units holds several advantages. Firstly, the edge-wise representations are agnostic to topologies and highly generalizable, enabling us to utilize simple cloth mesh for learning and directly apply it to various types of garments. Secondly, by conditioning $\phi_p(\cdot)$ on both the lengths and angles, EUNet can detect possible deformations and corresponding potential energy caused by different types of forces in a unified manner. For example, the changes in the edge length allow EUNet to approximate the stretching forces, while the angle variations enable EUNet to perceive the surface bending. Here, $\phi_d(\cdot)$ captures the air damping effects based on the velocities and scales according to the area of the adjacent faces, preventing the collapse of EUNet as demonstrated in the experiments. In addition, EUNet is translation- and rotation-equivalent as the inputs are invariant under different transformations.

To train our EUNet, we start with the approximation of the system's total energy between frames at $t + 1$ and $t$ as follows:

$$T_i^t = \frac{1}{2} m_i (\boldsymbol{v}_i^t)^\top \boldsymbol{v}_i^t, \quad V_{g,i}^t = -m_i \boldsymbol{g}^\top \boldsymbol{x}_i^t \tag{4}$$

$$\sum_i \left(T_i^t + V_{g,i}^t\right) + V_p^t = \sum_i \left(T_i^{t+1} + V_{g,i}^{t+1}\right) + V_p^{t+1} + V_d^{t+1}, \tag{5}$$

where $m_i$ and $\boldsymbol{v}_i^t$ are the mass and velocity of the vertex $i$, $\boldsymbol{g}$ is the gravity, $V_p$ and $V_d$ are the potential energy and dissipation energy, respectively. In discrete systems, we estimate the velocities

---

[0]https://www.blender.org/

by $\boldsymbol{v}_i^t = (\boldsymbol{x}_i^t - \boldsymbol{x}_i^{t-1})/h$. Due to the presence of damping effects, part of the energy possessed by the cloth at time $t$ is converted into the dissipation energy at $t+1$ as shown in Equation 5. Thus, we can extract the change of the cloth's energy as follows:

$$V_p^{t+1} + V_d^{t+1} - V_p^t \quad = - \sum_i \left(T_i^{t+1} + V_{g,i}^{t+1}\right) + \sum_i \left(T_i^t + V_{g,i}^t\right), \quad (6)$$

During training, we sample pairs of adjacent frames, namely $t+1$ and $t$, and apply square error to train our EUNet as follows:

$$\Delta V_\Phi \quad = \quad \Phi_p^{t+1} + \Phi_d^{t+1} - \Phi_p^t, \quad \Delta V = V_p^{t+1} + V_d^{t+1} - V_p^t, \quad (7)$$

$$\mathcal{L}_{\text{sup}} \quad = \quad (\Delta V_\Phi - \Delta V)^2. \quad (8)$$

**Vertex-Wise Contrastive Loss.** Through the approximation, we obtain system-wise scalars as supervision signals, which are the changes of potential energy and the dissipation energy. However, different combinations of deformations may result in similar energy potentials in total, leading to the loss of accuracy of EUNet. To improve the robustness and generalization ability of EUNet, we construct the vertex-wise contrastive loss through the analysis of the equilibrium of the system. According to backward-Euler commonly used in simulating clothes [9, 26, 10], the forward evolution of dynamics follows:

$$m_i \frac{\boldsymbol{x}_i - \boldsymbol{x}_i^t - h\boldsymbol{v}_i^t}{h^2} - m_i \boldsymbol{g} + \frac{\partial \Phi}{\partial \boldsymbol{x}_i} \quad = \quad 0. \quad (9)$$

Equivalently [9], the updates of the dynamics result from minimizing the constructed energy

$$K(\boldsymbol{x}_i, \hat{\boldsymbol{x}}_i) \quad = \quad \frac{1}{2h^2} m_i (\boldsymbol{x}_i - \hat{\boldsymbol{x}}_i^t)^\top (\boldsymbol{x}_i - \hat{\boldsymbol{x}}_i^t), \quad V_g(\boldsymbol{x}_i) = -m_i \boldsymbol{g}^\top \boldsymbol{x}_i \quad (10)$$

$$E(\boldsymbol{x}) \quad = \quad \sum_i K(\boldsymbol{x}_i, \hat{\boldsymbol{x}}_i) + \sum_i V_g(\boldsymbol{x}_i) + \Phi, \quad (11)$$

where $\hat{\boldsymbol{x}}_i^t = \boldsymbol{x}_i^t + h\boldsymbol{v}_i^t$. In other words, by assuming that the ground truth trajectory minimizes the system's energy, any disturbance in the system leads to an increase in total energy. Therefore, for the given trajectory $\boldsymbol{X}^t, \boldsymbol{X}^{t-1}$ and random disturbance $\Delta \boldsymbol{X}$, the system's energy becomes a function of our EUNet $\Phi$ and satisfies,

$$E(\Phi; \boldsymbol{X}^t + \Delta \boldsymbol{X}) - E(\Phi; \boldsymbol{X}^t) > 0. \quad (12)$$

To further deduce the vertex-wise relations, we expand the last term in Equation 9. Since the vertex normals are part of the inputs, the change of $\boldsymbol{x}_i$ affects the vertex normals of the nearest neighborhood vertices $j \in \mathcal{N}_i$, leading to energy variations in edges involving vertices from $\mathcal{N}_i$:

$$\frac{\partial \Phi}{\partial \boldsymbol{x}_i} \quad = \quad \sum_{j \in \mathcal{N}_i} \sum_{k \in \mathcal{N}_j} \frac{\partial \phi_{e_{jk}}}{\partial \boldsymbol{x}_i}. \quad (13)$$

More details about the energy variations can be found in the Appendix. Similarly, we construct the vertex-wise energy term so that minimizing the per-vertex energy is equivalent to solving Equation 9

$$E(\boldsymbol{x}_i) \quad = \quad K(\boldsymbol{x}_i, \hat{\boldsymbol{x}}_i) + V_g(\boldsymbol{x}_i) + \sum_{j \in \mathcal{N}_i} \sum_{k \in \mathcal{N}_j} \phi_{e_{jk}}, \quad (14)$$

where the disturbed set of vertices $i \in \mathcal{N}^*$ are selected so that the minimum distances between vertices are no less than four hops and the rest of the vertices remain unchanged. The per-vertex constraint given ground truth trajectory and disturbance becomes as follows:

$$E(\Phi; \boldsymbol{x}_i^t + \delta \boldsymbol{x}) - E(\Phi; \boldsymbol{x}_i^t) > 0, \quad i \in \mathcal{N}^* \quad (15)$$

During training, we apply the square error to constrain EUNet with Equation 15 by $\mathcal{L}_{\text{con}}$ and construct our final loss term $\mathcal{L}$ as follows:

$$\mathcal{L}_{\text{con}} \quad = \quad \sum_{i \in \mathcal{N}^*} \min(E(\Phi; \boldsymbol{x}_i^t + \delta \boldsymbol{x}) - E(\Phi; \boldsymbol{x}_i^t), 0)^2, \quad (16)$$

$$\mathcal{L} \quad = \quad \mathcal{L}_{\text{sup}} + \lambda \mathcal{L}_{con}, \quad (17)$$

where $\lambda$ is a hyperparameter to adjust the weight of vertex-wise contrastive loss.

Table 1: We report the mean and standard deviations of the square errors as indicated in Equation 8. We exhibit the impacts of our dissipation unit $\Phi_d$ and contrastive loss $\mathcal{L}_{\text{con}}$. Both of the components enable EUNet to ontain lower errors and smaller standard deviations, leading to higher accuracy and generalization ability.

| Configuration | Overall | Silk | Leather | Denim | Cotton |
|---|---|---|---|---|---|
| EUNet | **85.74±543.06** | **0.94±4.44** | **136.32±839.49** | **201.63±670.55** | 2.67±8.49 |
| EUNet w/o $\Phi_d$ | 211.76±1052.02 | 3.51±12.73 | 219.80±909.07 | 601.40±1819.94 | 14.06±36.50 |
| EUNet w/o $\mathcal{L}_{\text{con}}$ | 101.11±671.72 | 0.96±6.57 | 169.45±1020.94 | 230.34±852.87 | **2.38±6.31** |

(a) Potential energy mappings for silk        (b) Potential energy mappings for leather

Figure 3: Visualization of the potential energies predicted by our EUNet either without the dissipation energy branch $\Phi_d$ or the contrastive loss term $\mathcal{L}_{\text{con}}$. We sample the materials of silk and leather for demonstration, and change both the edge length and angles between vertex normals to verify the energy gradients caused by stretching and bending. Since silk is easier to bend, the energy gradients caused by different angles are smaller than those of leather. Both the dissipation energy branch and the contrastive loss enable EUNet to obtain reasonable energy gradients due to the deformations.

## 3.3 Dynamic Animation

Once our EUNet is trained, we obtain an energy function describing the constitutive relations together with the dissipation from the observed cloth in a unified manner. Following existing work [26, 10], we solve the dynamics of garments in a neural-network manner, where we apply our learned EUNet as a constrain through the loss term:

$$\mathcal{L}_{\text{dynamic}} = \mathcal{L}_{\text{inertia}} + \mathcal{L}_{\text{gravity}} + \mathcal{L}_{\text{collision}} + \mathcal{L}_{\text{external}} + \mathcal{L}_{\Phi}. \tag{18}$$

Specifically, $\mathcal{L}_{\text{external}} = -\sum_i \boldsymbol{f}_i^\top \boldsymbol{x}_i$ is the energy introduced by constant external force $\boldsymbol{f}_i$ applied to each vertex $i$, $\mathcal{L}_\Phi = \Phi$ is the energy predicted by our EUNet.

Since minimizing the total loss Equation 18 is equivalent to solving the motion equations constrained by backward Euler and our EUNet, the dynamic model can learn the garments' dynamics without any further data and simultaneously inherits the constitutive relations from the observed clothes. More details about the physics loss and the dynamic model can be found in the Appendix.

## 4 Experiments

### 4.1 Learning Constitutive Laws

As mentioned in Section 3.1, we generate 800 sequences of different clothes for training and 200 sequences for test, with a length of 30 frames for each sequence. The cloth is made of 4 types of materials typical in real life: silk, leather, denim, and cotton. The squared cloth is represented by mesh with 484 vertices and 1.4k edges. To build our EUNet, we apply four blocks of multi-layer perceptron (MLP) with dimensions 128. Each block of MLP consists of 2 fully connected layers. As for the contrastive loss in Equation 16, we apply randomly sampled noises from 10 normal distributions, whose standard deviation ranging from $\sigma/10$ to $\sigma$. Since the noises are small and only 10% of nodes are sampled for disturbance, the contrastive loss is relatively small. Thus, we set $\lambda = 10^6$ to ensure sufficient regularization. We adopt the Adam optimizer with an initial learning rate of 0.0002, with a decreasing factor of 0.5 every four epochs. The batch size is set to 4. We train our model for six epochs on V100.

We investigate the effectiveness of our EUNet and the impact of the dissipation energy unit $\Phi_d$ as well as the contrastive loss in Equation 16. Specifically, we evaluate the change of energy as indicated in Equation 8. The final quantitative results are the mean of errors from all adjacent frames as shown in Table 1. In addition, we further visualize the potential energy mappings predicted by

Table 2: Euclidean error (mm) on sampled Cloth3D [2] with sequence length of 80 frames. The garment-to-human collision rates are displayed under **Collision**. We use the garment-wise learning pipeline to train MGN [24] and LayersNet [28]. We further train a simulator based on MGN constrained by Saint Venant Kirchhoff (StVK) elastic model and the bending model as mentioned in [26], which is denoted by MGN-S+PHYS. In addition, based on the disentangled scheme, we combine our EUNet with energy optimization scheme and train models denoted by MGN-S+EUNet and MGN-H+EUNet, where MGN-H is the hierarchical graph neural network adopted in HOOD[10]. Simulators constrained by our EUNet achieve superior performance without access to the ground truth garments.

| Methods | Tshirt | Jumpsuit | Dress | Overall | Collision(%) |
|---|---|---|---|---|---|
| MGN [24] | 183.40±149.59 | 150.52±131.04 | 214.07±231.00 | 170.34±170.92 | 17.72±7.23 |
| LayersNet [28] | 114.74±97.54 | 80.96±27.69 | 108.06±69.74 | 92.72±60.93 | 14.61±6.71 |
| MGN-S+PHYS | 68.03±21.86 | 127.07±45.56 | 103.72±62.30 | 106.63±55.48 | 2.82±1.61 |
| HOOD [10] | 73.75±14.82 | 90.94±20.43 | **92.57±43.26** | 84.85±29.93 | 0.46±0.99 |
| MGN-S+EUNet (Ours) | **56.57±19.87** | 69.63±15.82 | 156.16±87.03 | 89.71±147.83 | **0.12±0.12** |
| MGN-H+EUNet (Ours) | 66.70±27.66 | **56.12±22.98** | 92.83±52.29 | **66.39±39.36** | 0.44±0.48 |

EUNet in Figure 3, indicating the constitutive relations resulting from stretching and bending. For the visualized potential energy unit, we vary the angles between vertex normals from -90 to 90 degree and increase the values of $\Delta l_e$ from -5mm to 5mm.

As shown in Table 1, our EUNet obtains low errors in predicting the cloth's energy with 1.4k energy units, suggesting the effectiveness of capturing the constitutive behaviors. Both the dissipation energy $\Phi_d$ and the contrastive loss $\mathcal{L}_{con}$ assist EUNet in reducing the quantitative errors.

Moreover, the visualized potential energy mappings in Figure 3 further demonstrate the accuracy of the predictions and the impacts of the two components. A reasonable potential energy mapping typically has the zero point in the rest state and shows an increase in energy with greater deformations. We sample two typical materials, silk and leather, for demonstration. Generally, since silk bends more easily than leather, the energy gradient of silk is smaller than that of leather. The dissipation energy unit is crucial in helping the model obtain reasonable gradients based on deformations, preventing the entanglement of damping effects with potential energy. For example, the energy increases as the bending angles increase. While the loss term $\mathcal{L}_{con}$ also leads to reasonable gradients, it is more effective in preventing model collapse. Both components contribute to generating reasonable constitutive relations, demonstrating the effectiveness and robustness of our EUNet.

## 4.2 Animating Garments

To further verify our EUNet and the disentangled scheme to learn garments, we evaluate the animations constrained by EUNet on ground truth garments from Cloth3D [2]. Since garments in Cloth3D are also made of silk, leather, denim, and cotton, the constitutive behaviors of the garments and our observed clothes are the same, leading to consistent dynamic deformations given external forces. Therefore, animations constrained by our EUNet should have similar dynamic sequences as the ground truth garments. We measure the Euclidean errors between the ground truth garments and the predictions for each frames, and report the mean and standard deviations of the errors. We also report the collision rates to demonstrate the plausibility of the predictions. The collision rates are obtained by calculating the number of penetrated vertices over the whole garment vertices. The errors for one sequence are averaged over frames. The final results are the mean of errors from all sequences.

As mentioned in Section 3.3, we obtain the garment animations constrained by EUNet through neural network-based simulators, which are MeshGraphNet (MGN-S) [24] and hierarchical graph neural network (MGN-H) adopted in HOOD [10]. We denote the simulator constrained by our EUNet as MGN-S+EUNet and MGN-H+EUNet in the following. We first compare models with garment-wise learning pipelines, which utilize the mean square errors and collision loss as mentioned in previous work [28] and train two baselines: MGN and LayersNet [28]. Notice that the MGN and MGN-S mentioned above have the same structures. Both MGN and LayersNet are trained using 50K frames of data from Cloth3D, with garments consist of 4k to 11k vertices. In addition, to compare the effectiveness of our EUNet with the analytical clothing model, we further train another simulator, which we denote by MGN-S+PHYS, constrained by the Saint Venant Kirchhoff (StVK) elastic model and the bending model as mentioned in [26]. Besides, we follow the work [10] to train the HOOD for comparisons. To ensure a fair comparison, we build a mini-network to estimate the material

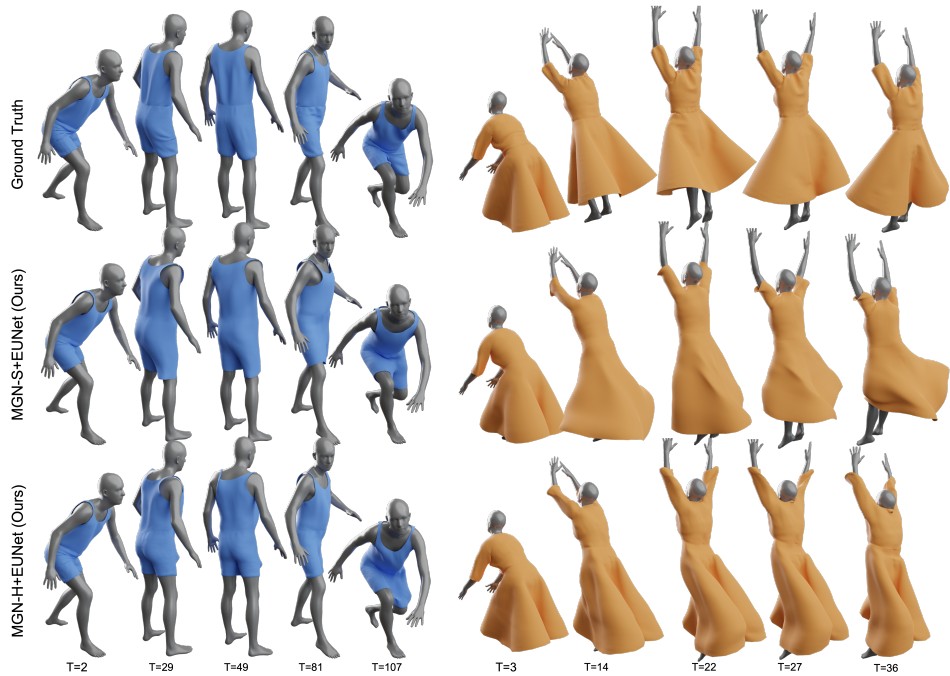

Figure 4: Qualitative results by our disentangled training scheme. We train MGN-S and MGN-H constrained by our EUNet through energy optimization scheme. Since the observed cloth to train EUNet is made of the same materials as the ground truth garments, the constitutive relations captured by EUNet are consistent with the ground truth data. As a result, MGN-S and MGN-H constrained by EUNet deliver similar deformation patterns as the ground truth garments without accessing any garment data. Even in long-term predictions, we can obtain plausible wrinkles, which are difficult for models trained in a garment-wise learning pipeline, and robust interactions with the human body.

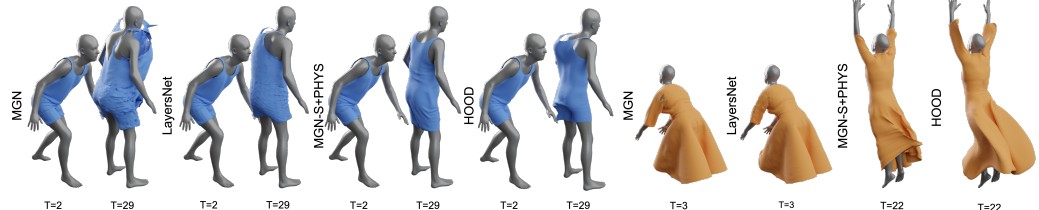

Figure 5: Qualitative results of garment animations by baselines. While the garment-wise learning scheme enables the baselines to obtain reasonable predictions within a short period of time, the errors increase for long-term predictions. Though we estimate the physics parameters required by the analytical clothing models, MGN+PHYS and HOOD generate overly soft garments and struggle to mimic the deformations of the ground truth data.

parameters required for the analytical clothing model to fit garments in Cloth3D, such as the Lame constants for the StVK elastic model. All models are tested using 10K frames of data from Cloth3D. We report the quantitative results in Table 5, and qualitative results in both Figure 4 and Figure 5. More details about the models can be found in the Appendix.

As shown in Table 5, our EUNet enables the MGN-S and MGN-H to accomplish superior performance compared to models trained through garment-wise learning in most cases, resulting in lower Euclidean errors and collision rates. Furthermore, MGN-S+EUNet and MGN-H+EUNet obtains higher accuracy than MGN-S+PHYS and HOOD constrained by the analytical clothing model, suggesting that our EUNet is more effective in capturing the observed constitutive laws than *fine-tuning the physics parameters* of analytical clothing model.

As shown in Figure 5, models trained garment-wise obtain faithful predictions in short sequences but tend to generate more artifacts as the sequence length increases due to the accumulated errors. Though the baselines are trained with the given ground truth data, they struggle to determine the relations between constitutive behaviors and dynamic laws. In contrast, as shown in Figure 4, MGN-S and

MGN-H constrained by EUNet achieve faithful deformations similar to the ground truth garments, suggesting the efficacy of our EUNet in capturing the constitutive relations. Moreover, since we adopt the disentangled training strategy, MGN-S+EUNet and MGN-H+EUNet are able to inherit the robustness from the energy optimization scheme and maintains stable and reasonable animations even for long-term predictions.

In addition, MGN-S+PHYS and HOOD, which are constrained by the analytical models, obtain overly soft behaviors of garments as shown in Figure 5. Though we estimate the physics parameters needed by the analytical models from the observed clothes, they struggles to deliver the constitutive behaviors of the ground truth garments, suggesting the gap between the analytical clothing model and the observed materials. On the contrary, MGN-S and MGN-H constrained by our EUNet achieve faithful animations with similar deformation patterns of the ground truth, indicating that our EUNet effectively captures the constitutive laws from the observations.

## 5  Conclusion

In this paper, we apply the disentangled scheme for learning garment animations into capturing constitutive relations from observations and dynamic animations based on energy optimizations. We deduce the variations of cloth energy from the trajectories to train our EUNet, which consist of potential energy unit and dissipation energy unit in topology independent design. To improve the accuracy and robustness of EUNet, we introduce disturbance to the sampled vertices in equilibrium state and constrain EUNet by vertex-wise contrastive loss. With the pre-trained EUNet, we can animate garments of diverse topologies based on the energy optimization, where EUNet serves as the material priors in the form of energy term. Consequently, we achieve more stable and physically plausible garment animations comparing with the baselines. The disentangled scheme alleviates the requirements of large scale of data, and combines the strong abilities of mimicking by EUNet  with robust capabilities of predicting dynamic evolution through energy optimization.

**Acknowledgement.** This study is supported under the RIE2020 Industry Alignment Fund Industry Collaboration Projects (IAF-ICP) Funding Initiative, as well as cash and in-kind contributions from the industry partner(s). It is also supported by Singapore MOE AcRF Tier 2 (MOE-T2EP20221-0011) and partially funded by the Shanghai Artificial Intelligence Laboratory.

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

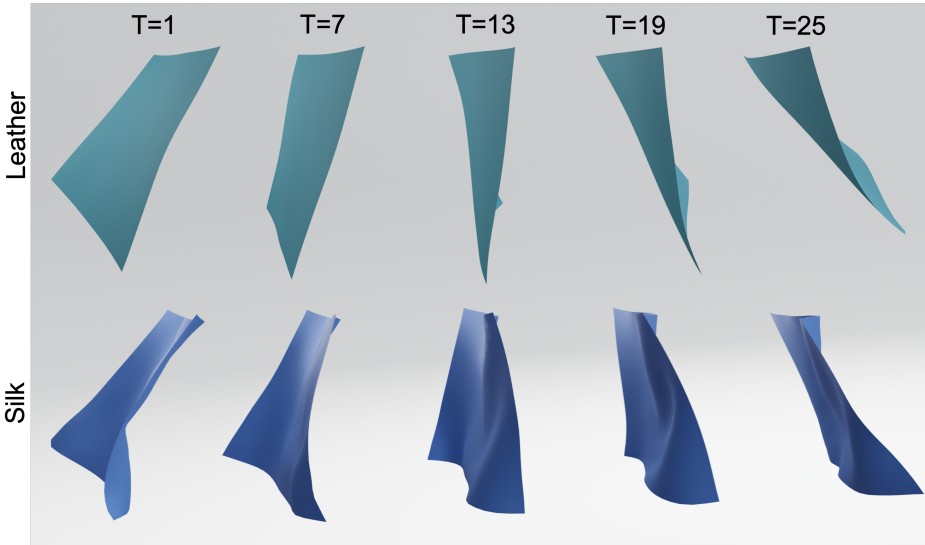

Figure 6: The observed trajectories of cloth for training EUNet. The cloth is made of different materials and pinned by two corners on top. We randomly initialize the positions with different velocities for the cloth and let the cloth deform under the influence of forces.

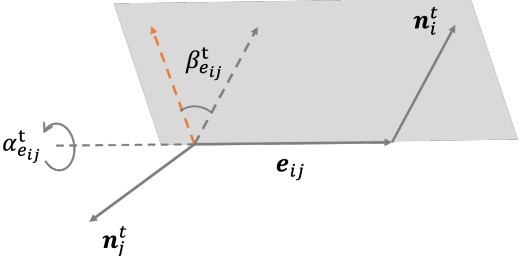

Figure 7: We represent the angles between vertex normals $\boldsymbol{n}_i^t$ and $\boldsymbol{n}_j^t$ by $[\alpha_{e_{ij}}^t, \beta_{e_{ij}}^t]$, where $\alpha_{e_{ij}}^t$ is the rotation angle along edge vector $\boldsymbol{e}_{ij}$ from $\boldsymbol{n}_j^t$ to the plane defined by $\boldsymbol{n}_i^t$ and $\boldsymbol{e}_{ij}$, $\beta_{e_{ij}}^t$ is the angle between $\boldsymbol{n}_i^t$ and the rotated vertex normal within the plane.

# A Appendix

## A.1 Methodology

### A.1.1 Data for Training EUNet

We select cloth made of typical materials in real-life: silk, leather, denim, and cotton. The square cloth is pinned at the two corners on top as shown in Figure 6. The data are simulated by Blender[1] in the format of mesh, which are represented by 484 vertices and 1.4K edges. The topology for the cloth is the same for all trajectories. We generate 800 sequences for training and 200 sequences for test. Each sequence include 30 frames of cloth.

### A.1.2 Edge-Wise Potential Energy $\Phi_p(\cdot)$

As shown in Equation 2, the function takes the length differences $\Delta l_{e_{ij}}^t \in \mathbb{R}$, the relative angles $\Delta \boldsymbol{\theta}_{e_{ij}}^t \in \mathbb{R}^2$, and the attribute vectors $\boldsymbol{a} \in \mathbb{R}^5$ as inputs. Specifically, as shown in Figure 7 to reduce the ambiguity, we represent the angles $\boldsymbol{\theta}_{e_{ij}}^t$ between two vectors in 3D space by two separate angles $[\alpha_{e_{ij}}^t, \beta_{e_{ij}}^t]$. For edge vector $\boldsymbol{e}_{ij}$ and vertex normals $\boldsymbol{n}_i^t$ and $\boldsymbol{n}_j^t$, $\alpha_{e_{ij}}^t$ is the rotation angle along $\boldsymbol{e}_{ij}$ from $\boldsymbol{n}_j^t$ to the plane defined by $\boldsymbol{n}_i^t$ and $\boldsymbol{e}_{ij}$, $\beta_{e_{ij}}^t$ is the angle between $\boldsymbol{n}_i^t$ and the rotated vertex normal within the plane.

---

[1] https://www.blender.org/

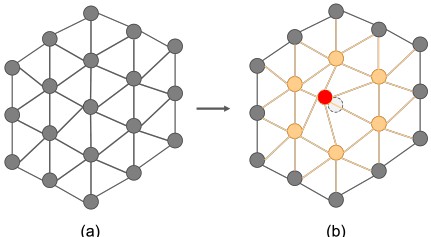

(a)           (b)

Figure 8: Demonstration of the impact caused by the disturbed vertex. Suppose figure (a) is the ground truth position for each vertex at time $t+1$. When adding noise to the vertex $i$ at the center, the vertex normals of the surrounding orange vertices $j \in \mathcal{N}_i$ are disturbed. As a result, energy units for edges, which connect to both vertex $i$ and vertices $j \in \mathcal{N}_i$ and are marked by orange, change accordingly.

### A.1.3 Vertex-Wise Contrastive Loss

Here we explain how the change of one node affects the corresponding edges. The change of the position $x_i$ affects the vertex normals of the nearest neighborhood vertices $j \in \mathcal{N}_i$, as denoted by the orange nodes in Figure 8. Consequently, the energy for edges, marked by orange in Figure 8, is also changed.

### A.1.4 Loss Terms for the Energy Optimization

As mentioned in the main text, the total loss to train a model constrained by our EUNet is as follows:

$$\mathcal{L}_{\text{dynamic}} = \mathcal{L}_{\text{inertia}} + \mathcal{L}_{\text{gravity}} + \mathcal{L}_{\text{collision}} + \mathcal{L}_{\text{external}} + \mathcal{L}_{\Phi}, \tag{19}$$

where $\mathcal{L}_{\Phi} = \Phi$ is the predicted energy by our EUNet. Specifically, $\mathcal{L}_{\text{inertia}}$ and $\mathcal{L}_{\text{gravity}}$ are the inertia and gravity energies:

$$\hat{x}_i^t = x_i^t + h v_i^t \tag{20}$$

$$\mathcal{L}_{\text{inertia}}(x, \hat{x}^t) = \sum_i \frac{1}{2h^2} m_i (x_i - \hat{x}_i^t)^\top (x_i - \hat{x}_i^t) \tag{21}$$

$$\mathcal{L}_{\text{gravity}}(x) = -\sum_i m_i g^\top x_i, \tag{22}$$

where $h$ is the time interval between adjacent frames. The collision term aims to reduce the penetrations between garments and human bodies and has a similar format as existing work [26, 10]:

$$d(x_i, x_o) = (x_i - x_o) \cdot n_o \tag{23}$$

$$\mathcal{L}_{\text{collision}} = \max(\epsilon - d(x_i, x_o), 0)^3, \tag{24}$$

where $x_o$ is the position of the $o$-th face center from the human mesh at time $t+1$ and $n_h$ is the face normal. $\epsilon$ is a safe boundary and is set as 0.002 in practice. The index of $o$ indicates the closest face center to the vertex $x_i^t$ within a range of 0.03mm at time $t$. Additionally, we randomly apply constant external forces $f$ besides the gravity to each vertex and penalize the corresponding energies:

$$\mathcal{L}_{\text{external}}(x_i) = -\sum_i f_i^\top x_i. \tag{25}$$

## A.2 Experiments

### A.2.1 Implementation Details for Models

**MeshGraphNet (MGN) [24].** The MGNs used for the disentangled scheme and garment-wise training pipelines are the same. Specifically, the MGN is formatted as follows:

$$\hat{\alpha}^{t+1} = F(\bar{M}, M_h^t, M_h^{t+1}, a, g, \{\frac{f_i}{m_i}\}_{i=0}^N), \tag{26}$$

where $\alpha^{t+1}$ are the predicted accelerations for each vertex at time $t+1$, $\bar{M} = \{T, E\}$ represents the garment template mesh with corresponding vertices $T$ and $E$ edges. $M_o^t = \{V_o^t, E_o\}$ and

Table 3: Model configurations. We mark the MGN and LayersNet trained in supervised manner by "Supervised". We denote the HOOD's hierarchical graph neural network by MGN-H. The "Friction" loss term from HOOD introduces the frictions between human body and garments.

| Methods | Neural Network Structure | Self-Supervised Loss Components | | |
| --- | --- | --- | --- | --- |
| | | PHYS | Friction | EUNet (ours) |
| MGN (Supervised) | MGN | | | |
| LayersNet (Supervised) | LayersNet | | | |
| MGN-S+PHYS | MGN | ✓ | | |
| MGN-S+EUNet | MGN | | | ✓ |
| MGN-S+PHYS+F | MGN | ✓ | ✓ | |
| MGN-H+PHYS+F (**HOOD**) | MGN-H | ✓ | ✓ | |
| MGN-H+EUNet | MGN-H | | | ✓ |

$M_o^{t+1} = \{\boldsymbol{V}_o^{t+1}, \boldsymbol{E}_o\}$ are the meshes for the underlying human bodies at time $t$ and $t+1$ respectively. The predicted positions for garment vertices at time $t+1$ are calculated as follows:

$$\hat{\boldsymbol{v}}^{t+1} = \boldsymbol{v}^t + h\hat{\boldsymbol{\alpha}}^{t+1} \tag{27}$$
$$\hat{\boldsymbol{x}}^{t+1} = \boldsymbol{x}^t + h\hat{\boldsymbol{v}}^{t+1}. \tag{28}$$

The input for the MGN has similar format as [10]. In addition, we concatenate the accelerations caused by constant forces as part of the vertex feature.

$$\boldsymbol{\alpha}_i^{\text{external}} = \boldsymbol{g} + \frac{\boldsymbol{f}_i}{m_i} \tag{29}$$

As for the MGN-S constrained by analytical clothing model, we build a mini-network to approximate the physics parameters from the observed garments in Cloth3d to fit the clothing model.

$$m, \mu, \lambda, k = F(\hat{m}, \boldsymbol{a}), \tag{30}$$

where $m$ is the mass for the analytical clothing model. $\mu$ and $\lambda$ are the lame constants for the StVK elastic model. $k$ is the constant for the bending model. $\hat{m}$ is the mass from the Cloth3D. $\boldsymbol{a}$ is the attribute vector mentioned in Cloth3D, such as the tension stiffness, bending stiffness, and the tightness. All the input and output attributes are normalized according to the minimum and maximum values. The mini-network consist of 3 layers of multi-layer perceptrons with hidden dimension 64. We train the mini-network with initial learning rate 0.0002 and batch size 4 for 4 epochs. We sample 10k frames from Cloth3D for training. We adopt Adam optimizer with decreasing factor of 0.5 for each epoch.

**LayersNet [28].** We followed the official work [28] to train the model on Cloth3D dataset. Specifically, we adopt the mean square errors as loss term to penalize the differences between the predicted positions and ground truth data. We also apply the collision loss mentioned in the work [28] to solve the collision between human body. Since we do not consider multiple layers of garments, we only penalize the collisions between the single-layered garments and human body mesh.

All models are trained on V100.

### A.2.2 Experiment Results of Garment Animation

In this section, we report extra quantitative and qualitative results in Table 4 and Figure 9. We summarize the models architectures and the loss terms used in training process in Table 3.

As shown in Table 4, simulators constrained by our EUNet deliver superior performance than baselines, suggesting the effectiveness of our EUNet.

In addition, we demonstrate the effectiveness of the bending forces captured by our EUNetas shown in Figure 9 Specifically, we force EUNet not to generate energy gradients when bending, and train a MGN as simulator. Without the bending forces, the animations tend to generate more wrinkles but different deformations comparing with the ground truth data.

Table 4: Euclidean error (mm) on sampled Cloth3D with sequence length of 80 frames. Comparing "MGN-S+PHYS" with "MGN-S+PHYS+F", the friction loss term does not significantly affect the overall performance. Comparing "HOOD w/o EST", which does not use the mini-network to estimate the material parameters as mentioned in Section 4.2, with "HOOD", the mini-network is necessary to improve the accuracy for simulators constrained by analytical clothing model. Comparing "HOOD" with "MGN-H+EUNet (Ours)" and "MGN-S+PHYS" with "MGN-S+EUNet (Ours)", simulators constrained by our EUNet achieves lower errors, suggesting the effectiveness of EUNet.

| Methods | Tshirt | Jumpsuit | Dress | Overall | Collision(%) |
|---|---|---|---|---|---|
| MGN [24] | 183.40±149.59 | 150.52±131.04 | 214.07±231.00 | 170.34±170.92 | 17.72±7.23 |
| LayersNet [28] | 114.74±97.54 | 80.96±27.69 | 108.06±69.74 | 92.72±60.93 | 14.61±6.71 |
| MGN-S+PHYS | 68.03±21.86 | 127.07±45.56 | 103.72±62.30 | 106.63±55.48 | 2.82±1.61 |
| MGN-S+PHYS+F | 76.58±19.38 | 130.42±45.76 | 101.47±41.27 | 108.83±45.55 | 2.92±1.48 |
| HOOD w/o EST | 111.27±22.84 | 182.89±66.92 | 163.90±64.87 | 153.78±63.75 | 0.36±0.27 |
| HOOD [10] | 73.75±14.82 | 90.94±20.43 | **92.57±43.26** | 84.85±29.93 | 0.46±0.99 |
| MGN-S+EUNet (Ours) | **56.57±19.87** | 69.63±15.82 | 156.16±87.03 | 89.71±147.83 | **0.12±0.12** |
| MGN-H+EUNet (Ours) | 66.70±27.66 | **56.12±22.98** | 92.83±52.29 | **66.39±39.36** | 0.44±0.48 |

Table 5: Time efficiency comparisons between analytical models and our EUNet. We denote the StVK elastic model and the bending model by "PHYS", which is used in the main text. As formulated in Equation 1, our EUNet is composed of two separate branches: $\Phi_p$ for potential energy and $\Phi_d$ for dissipation. Both branches have the same structures. We report the time separately for each branch and the full EUNet as follows. The forward time is averaged on 80 frames of predictions, which include garments composed of 7924 vertices, 23636 edges and 15712 faces. All experiments are run on NVIDIA A100-SXM4-80GB.

| | PHYS | EUNet $\Phi_p$ | EUNet $\Phi_d$ | EUNet $\Phi_p$+$\Phi_d$ |
|---|---|---|---|---|
| Time (ms) | 1.743+0.212 | 1.055+0.202 | 1.216+0.223 | 2.271+0.422 |

### A.2.3 Time Efficiency Comparisons

We report the time efficiency in Table 5. We compare the analytical models, which are the StVK elastic model and bending model adopted by the baselines, and our EUNet. As formulated in Equation 1, our EUNet composed of two separate branches: $\Phi_p$ for potential energy and $\Phi_d$ for dissipation. Both branches have the same structures. We report the time separately for each branch and the full EUNet as follows. The forward time is averaged on 80 frames of predictions, which include garments composed of 7924 vertices, 23636 edges and 15712 faces. All experiments are run on NVIDIA A100-SXM4-80GB.

As shown in Table 5, our EUNet is comparable to the traditional clothing models in terms of speed.

### A.3 Limitations

A possible limitation is that our EUNet is based on the edge-wise discretization of the continuous mesh. While polygonal discretization would potentially improve the performance, we can also construct the polygon-wise energy unit by the summation of the edge-wise units. Another limitation is that our EUNet does not consider the self-collisions during training. Since we do not require the trajectories needed to train EUNet to be very complex, we can adopt and generate simple cloth motions, such as the pinned cloth falling under the influence of gravity, to avoid the self-collisions in the training data. As demonstrated in our work, our EUNet effectively captures the constitutive behaviors with the simple motions of the cloth.

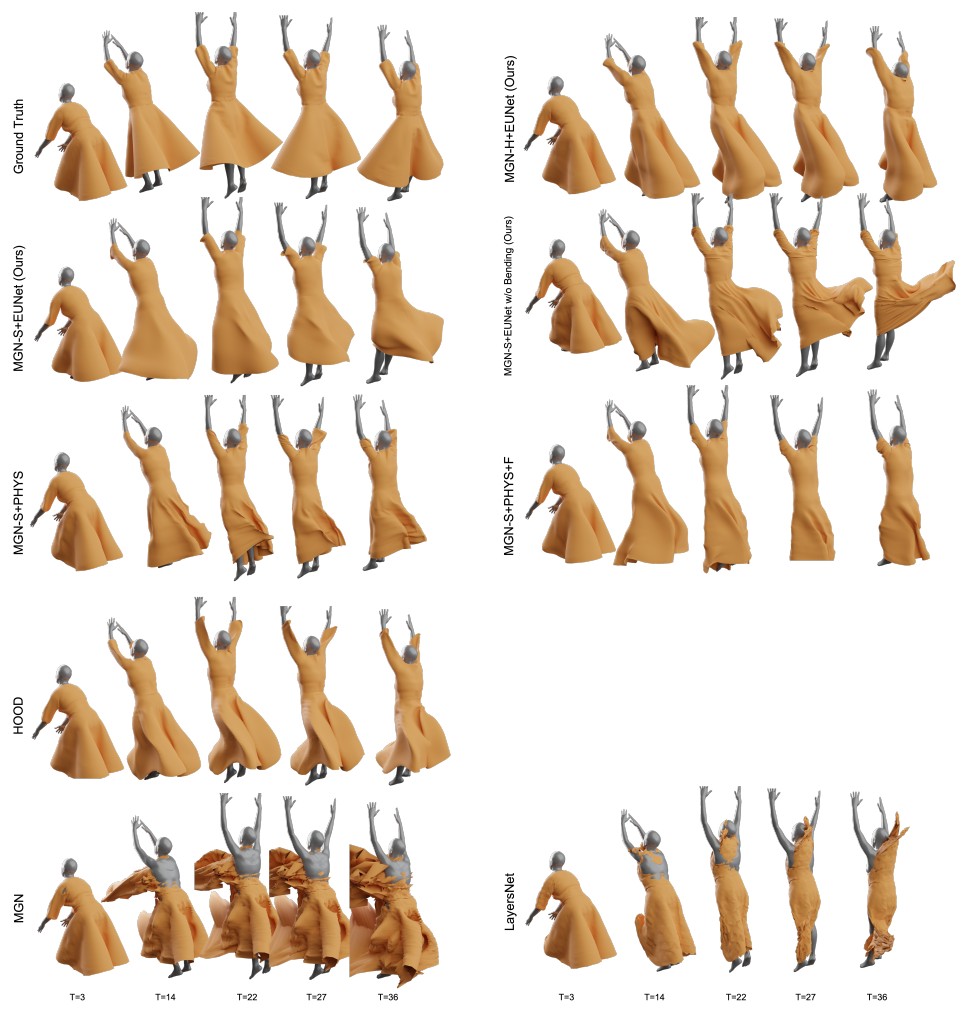

Figure 9: We display the qualitative results for all models. Simulators constrained by our EUNet deliver dynamic patterns closer to the ground truth. We further demonstrate the effectiveness of the bending forces captured by our EUNet, which is shown on the second row. Without the bending forces, the MGN-S+EUNet tends to generate more wrinkles but different deformation patterns as the ground truth garments.

