# OpenReview forum: "Learning 3D Garment Animation from Trajectories of A Piece of Cloth"
_NeurIPS.cc/2024/Conference — NeurIPS 2024 poster_

### Official Review · Reviewer_cPXz · 2024-06-12

**Soundness:** 2
**Presentation:** 3
**Contribution:** 2
**Rating:** 4
**Confidence:** 4

**Summary:**

The authors propose a method to transfer the deformations of the observed garments to any other garment. Previous methods either rely on a large-scale dataset for training or analytical physics model with limited expressive ability. On the contrast, the proposed method first learns the constitutive relations from the observation by a neural network (EUNet), then use it as an energy prior to regularize the training of garment deformation model. This design addresses the limitation of previous works and show better results.

**Strengths:**

The strength of the paper is the proposed method does not need to collect huge amount of data with varied body pose and shape and garment types for training. Through theoretical analysis, they prove that they can learn a more physically accurate energy model to describe the deformation of garment. In this way, they do not need an explicit physical model, which tends to have limited expressive power. The derivation is theoretical sound.

**Weaknesses:**

To learn this energy model by EUNet, the authors rely on the synthetic data simulated with blender. A cloth with known geometry (vertices and faces) is assigned with a specific material type. However, this setting is too ideal. In real scenarios, we are more interested in transferring the material of a real cloth to another garment. But having the geometry of a real cloth usually is not feasible. Even though we can have the mesh of the cloth through some registration process, how to get the shape of mesh when it is hanging and dangling is still a problem. In this paper, I do not see the possibility of using the proposed method in real applications. This is the critical weakness.

**Questions:**

1. In Fig. 4, the results of MGN-S+EUNet on dress are not similar to the ground truth data.
2. What is the unit of the errors in Table 1? The errors on the leather and denim seem too big compared with the others.

**Limitations:**

See the weakness.

---

> ### Author Rebuttal · Authors · 2024-08-07
>
> We thank the reviewer for their valuable feedback.
>
> # W1: Possibility of Using The Method in Real Applications
> Firstly,
> synthetic data is commonly used and facilitate the research of garment animations,
> such as TailorsNet [1], Cloth3D [2], MotionGuided [3], Clo3D [4].
>
> Secondly,
> as shown in the dataset [5],
> it is feasible to obtain temporally consistent cloth data represented by vertices from real life, which can be directly applied by our method to learn the constitutive laws in real world, though it is not yet publicly available.
>
> Thirdly,
> the garment geometry can be extracted either through learning based methods [6, 7] or from raw scans [8],
> making it possible to animate realistic garments.
> As shown in Figure 2 in additional PDF from global rebuttal,
> we animate the garments from real scans [8] for demonstration,
> transferring the virtual material from Blender to real-life garments from CAPE [8].
>
> In conclusion,
> our work can be directly applied to real-life applications related to garment animations.
>
>
> [1] Chaitanya Patel, et al. TailorNet: Predicting Clothing in 3D as a Function of Human Pose, Shape and Garment Style. CVPR2020
>
> [2] Hugo Bertiche, et al. CLOTH3D: Clothed 3D Humans. ECCV2020
>
> [3] Meng Zhang, et al.  Motion guided deep dynamic 3D garments. TOG2022
>
> [4] Xingxing Zou, et al. Cloth4d: A dataset for clothed human reconstruction. CVPR2023
>
> [5] David Blanco Mulero, et al. Benchmarking the Sim-to-Real Gap in Cloth Manipulation. IEEE Robotics Autom2024
>
> [6] Heming Zhu, et al. Registering Explicit to Implicit: Towards High-Fidelity Garment mesh Reconstruction from Single. CVPR2022.
>
> [7] Lingteng Qiu, et al. REC-MV: REconstructing 3D Dynamic Cloth from Monocular Videos. CVPR2023
>
> [8] Qianli Ma, et al. Learning to Dress 3D People in Generative Clothing. CVPR2020
>
> # Q1: Qualitative Results of Dress
> We acknowledge that predicting long-term dynamics in an auto-regressive manner inevitably leads to error accumulation.
> The baseline models either struggle with predicting long-term animations or fail to replicate similar constitutive behaviors observed in the ground truth.
>
> In contrast,
> MGN-S+EUNet (ours) delivers material behaviors that more closely match the ground truth,
> such as a more rigid dress in Figure 4 of the main manuscript,
> and achieves lower prediction errors as shown in Table 2 of the main manuscript,
> suggesting the effectiveness of our learned EUNet.
> Please refer to the video in supplementary for better comparisons
> and the additional PDF file for more comparisons.
>
> # Q2: Explanations of Errors in Table 1 of The Main Manuscript
> The unit is the energy $kg\cdot m^2/s^2$.
> The errors in Table 1 of the main manuscript are the square errors as indicated in Equation 8,
> where we sum up 1.4K energy units of the cloth with 484 vertices.
> The per-energy-unit errors are significantly smaller in terms of the absolute value.
>
> Regarding the differences between the materials,
> leather and denim are either stiffer or heavier.
> These characteristics lead to larger energy changes between frames due to significant variations in velocity or increased mass,
> making it more challenging for the model to learn.
> We would include the discussion in the limitation section in our revised version.
>
> In conclusion,
> as shown in Table 2 of the main manuscript,
> MGN-S+EUNet (ours) still achieves superior performance,
> suggesting the effectiveness of our EUNet.
> And as shown in Figure 1 of the main manuscript and the video in supplementary,
> MGN-S+EUNet delivers reasonable material behaviors for garments made of silk and leather,
> which are softer or stiffer respectively.

---

> > ### Comment · Reviewer_cPXz · 2024-08-10
> >
> > Thanks for your clarification and efforts in the rebuttal.
> >
> > I agree that many synthetic data is widely used for garment animation or simulation. However, the simulated data is just approximation for the real data. As pointed by David Blanco Mulero, et al [5], there is a large gap between the synthetic data and the real data. Besides, the temporal data collected in [5] are point clouds instead of meshes with valid geometry. The proposed method requires the trajectory of mesh as input. I do not think the real data of [5] can be used by it to learn the constitutive laws.

---

> > > ### Author Response · Authors · 2024-08-11
> > > **Official Comment by Authors**
> > >
> > > We thank the reviewer for the response.
> > > We first clarify the potential use of dataset [1],
> > > followed by a more explicit summary of our rebuttal addressing the concerns raised in the weaknesses.
> > >
> > > Firstly,
> > > many existing softwares,
> > > such as MeshLab and AutoDesk ReCap,
> > > support the generation of meshes from point clouds.
> > > Therefore,
> > > a straightforward solution is to preprocess the point clouds from the dataset [1] using these tools and extract the meshes in the format required by our inputs.
> > > Many work,
> > > such as [2],
> > > have developed techniques to generate meshes from point clouds.
> > > And since the vertices are temporally consistent,
> > > the trajectories of the meshes remain consistent as well.
> > >
> > > Secondly,
> > > we believe the concerns in weaknesses result from **potential difficulties in collecting real-world data**,
> > > as mentioned by "...A cloth with known geometry...The setting is too ideal...having the geometry of a real cloth usually is not feasible..." and "...how to get the shape of mesh when it is hanging and dangling is still a problem...",
> > > leading to the key concern "...do not see the possibility of using the proposed method in real applications...".
> > >
> > > However,
> > > the **dataset [1] from real-world closely resembles our settings**
> > > and captures temporally consistent data **when the cloth is "hanging and dangling"**.
> > > As mentioned in the previous rebuttal,
> > > the garment geometry can be extracted either through learning based methods or real scans.
> > > Examples from CAPE [3] even include **meshes of clothed humans obtained from real scans**.
> > > The animaitons of garments from CAPE,
> > > as demonstrated in the rebuttal,
> > > showcase the **real application of garment animation**.
> > > In other words,
> > > collecting real data in the format required by our method is feasible, making it applicable for real-world applications
> > >
> > > In addition,
> > > while our work is trained and validated using synthetic data,
> > > which is agreed by the reviewer,
> > > our EUNet is also applicable to real data.
> > > The key assumption is that **the dynamics of objects follow general physical laws**,
> > > such as Newton's laws or, equivalently, the Lagrangian mechanics, which apply to both simulated and real-world data.
> > > The primary difference is that the clothing model in simulations is typically known, whereas in real life, it is unknown.
> > > Our EUNet is designed to model these underlying constitutive behaviors.
> > > Therefore, our approach can be directly applied to real data without any modification.
> > > Together with the animations of real garment in CAPE,
> > > both our core module, EUNet, and the garment animation method **demonstrate the potential for real-world applications**.
> > >
> > > We hope that the clarification provided above addresses the reviewer's concerns more explicitly,
> > > including "...do not think the real data can be used...", "...The setting is too ideal...having the geometry of a real cloth usually is not feasible...", "...how to get the shape of mesh when it is hanging and dangling is still a problem...",
> > > and "...do not see the possibility of using the proposed method in real applications...".
> > >
> > > [1] David Blanco Mulero, et al. Benchmarking the Sim-to-Real Gap in Cloth Manipulation. IEEE Robotics Autom2024
> > >
> > > [2] Rana Hanocka, et al. Point2Mesh: A Self-Prior for Deformable Meshes. TOG2020
> > >
> > > [3] Qianli Ma, et al. Learning to Dress 3D People in Generative Clothing. CVPR2020

---

### Official Review · Reviewer_CDgw · 2024-07-10

**Soundness:** 3
**Presentation:** 3
**Contribution:** 3
**Rating:** 6
**Confidence:** 2

**Summary:**

This submission presents a method that could effectively learn the dynamic patterns of different garments from a single piece of cloth. The key insight is that the motion of different cloths is governed by both external forces and the constitutive relations rather than specific garment topologies. Thus, an Energy Unit Network (EUNet) is proposed to learn the topology independent constitutive laws. Then the animation is performed by energy optimizations. Experimental results shows improved results comparing to previous methods and baseline methods.

**Strengths:**

The paper is well written and easy to read.
1)	The paper is well structured.
2)	Many terms are well defined and explained.
The proposed method is both novel and interesting.
1)	The disentangled learning scheme of using a network to learn the constitutive law, which generalizes to different garment types, is physically intuitive and natural. More importantly, this design helps alleviate the needs for large amount of training data of various cloth shapes in dynamic for learning based animation.  This disentanglement between topology and energy is achieved by using mesh edge as a unit instead of the whole cloth mesh.
2)	The proposed disturbance training strategy helps stabilize the training and improves the generalization of EUNet. As a constraint, it accompanies the direct supervision on the energy form by taking into account the physical meaning of equilibrium state. This helps the network to learn a more reasonable manifold of the energy distribution.
Experimental results:
1)	Improved results over previous methods are shown both qualitatively and quantitatively.
2)	According to the ablation study, the design of both the contrastive loss and dissipation unit loss are validated as effective.

**Weaknesses:**

Some details regarding the design of the EUNet is missing. Although some descripsions on the EUNet design is provided at the experimental section, it is relatively hard for the reader to follow and develop a more coherent understanding of the presented work.

Some limitations and open questions that the work might not cover:
What about anisotropic materials? How to adapt the current model design to also fit to cloths where its material is anisotropic? How well does the method handles cloths with more complex topology that goes beyond a single layer of cloth? As also pointed by the authors, the method does not handle self-collision. It would be interesting to see how it can be adapted in that axis.

**Questions:**

See weakness.

**Limitations:**

See weakness.

---

> ### Author Rebuttal · Authors · 2024-08-07
>
> We thank the reviewer for the positive feedback
> and recognizing the value of our work.
> We believe the reviewer has sufficient understandings of our framework and pipeline.
>
> # W1: Details Regarding The Design of EUNet
> We introduce the formulations and training procedures of our EUNet in Section 3.2,
> and include the implementation details in Section 4.1 at L239-249.
>
> Furthermore,
> we report the time efficiency of our EUNet in the global rebuttal.
>
> Please let us know if the reviewer still has any confusion about our EUNet.
> We will clarify it further in our revised version to ensure it is easier for readers to follow.
>
> # W2.1: Anisotropic Materials
> A naive extension is to add the directional information as extra inputs for our EUNet.
> As a simplified example,
> we assume a rectangular wooden board that is thin enough for its thickness to be negligible.
> The board is stiffer along the x-axis than the y-axis.
> We then discretize the thin board by edges along both the x-axis and y-axis.
> To model such material,
> we can extend our EUNet by taking the edges' directions,
> which are either x-axis or y-axis in this example,
> as extra inputs by $\Phi(\cdots, \mathbf{d})$,
> where $\mathbf{d}$ indicates the directional information.
>
> # W2.2: More Complex Topology Than A Piece of Cloth
> For cloth or garments with complex topologies used as training data,
> our EUNet can directly learn their constitutive behaviors thanks to the topology-independent design.
> To achieve accurate results,
> it is essential to ensure that the training sequences involve minimal collisions,
> so that the dynamics and deformations are primarily caused by internal forces closely related to potential energy.
>
> The reason we chose to use a simple piece of cloth as training data is that this setup is easier to achieve with high quality in real-life scenarios.
> For example,
> [1] provides the temporally consistent cloth data represented by vertices from real life,
> which can be directly applied by our EUNet but not publicly available yet.
>
> [1] David Blanco Mulero, et al. Benchmarking the Sim-to-Real Gap in Cloth Manipulation. IEEE Robotics Autom2024
>
> # W2.3: Handling of Self-collisions
> Since collisions are independent of the internal forces caused by deformation,
> the key to accurately modeling potential energy is to minimize the effects of collisions.
> However,
> to extend our method to more complex scenarios involving collisions,
> one could design a new module that takes the distances between edges or faces of the cloth as inputs
> to model the dissipation energy caused by collisions.
>
> In this work,
> we minimize the effects of collisions during the data generation process.
> We will include such scenarios in future work.

---

### Official Review · Reviewer_QaVe · 2024-07-11

**Soundness:** 2
**Presentation:** 3
**Contribution:** 3
**Rating:** 5
**Confidence:** 4

**Summary:**

This work proposes a method to learn the constitutive model of cloth materials from observed cloth trajectory using a neural network. It adopts an MLP that operates on individual edges and predicts per-edge distortion based on the deviation of edge geometry from rest shape and trains the network using a combination of supervision on potential energy change with ground truth and optimality of incremental potential. The learned potential energy can be used as a constraint to train neural simulators for garment animation.

**Strengths:**

- I appreciate the novelty in the idea of learning the constitutive model of cloth materials in a data-driven manner. Potentially this formulation could allow the neural networks to understand the intrinsic physical property instead of mimicking the behavior of specific examples, thus of scientific significance if implemented correctly.
- The paper is well-written and mostly clear.

**Weaknesses:**

- On the methodology side, the major question is probably the design of dissipative energy. On the one hand, why it is and is merely a function of \(X^t - X^{t-1}\) is questionable. In fact, whether it should be modeled as an absolute quantity is a question because the total amount of dissipative energy seems not that meaningful. The only observable quantity is the relative change of dissipative energy in a physical process.
- On the other hand, with the presented framework, it is very hard to learn the major sources of energy dissipation: collision, and friction, since they are neither present in the training data, nor fully modeled (e.g. self-collision) in the formulation. While the dissipative energy is not the focus, the problem is that without correctly modeling dissipative forces, I doubt the possibility of learning an accurate elastic potential energy function.
- On the evaluation side, the problem is that the method is only evaluated in a simplified setting, without comparing against methods or in settings that are practically useful (see below). In my opinion, there are two ways to demonstrate that the learned constitutive model is useful: either 1. demonstrate that it is more accurate than an analytical model on real data, or 2. show that it leads to more realistic animation than existing methods (including traditional numerical models).
- The evaluation section only shows that the MGN trained with the learned constitutive model is better than those trained with ground truth garments or analytical constitutive model. On the one hand, it does not compare with other state-of-the-arts like HOOD, SNUG, and PBNS that are also formulated in a self-supervised manner. On the other hand, the claim that it is better than the analytical constitutive model is not convincing because the discrepancy may be caused by the limited accuracy in the neural simulator (or even by the mini-network mentioned in Sec 3.3). To truly demonstrate that it is better than an analytical one, it must be compared using a numerical integrator that is guaranteed to converge to the energy minimum.

**Questions:**

See the weaknesses section.

**Limitations:**

The discussion seems adequate.

---

> ### Author Rebuttal · Authors · 2024-08-07
>
> We thank the reviewer for their valuable feedback.
>
> # W1: Design of Dissipative Energy
> As commonly adopted in physics simulation,
> such as the Rayleigh dissipation,
> the dissipation can be approximated by a function of objects' velocities,
> which can be calculated from $X^t-X^{t-1}$ in our work.
>
> In addition,
> we agree that the absolute value of the dissipative energy is not meaningful.
> However,
> during the energy optimization process of solving dynamics,
> only the derivative of the dissipation energy $\lim_{\Delta x \to 0}\frac{\Phi_d(x+\Delta x, \cdot)-\Phi_d(x, \cdot)}{\Delta x}$ will take effect,
> and eliminate the potential impacts from the absolute values.
>
> Lastly,
> if there exists a constant $C>0$ in the output of our $\Phi_d$,
> it indicates that the system from Blender has a constant damping effect.
> When the velocity is 0, the output of $\Phi_d$ is $C=3\times 10^{-6}$, which is small enough to be negligible.
>
> # W2: Dealing with Collision And Friction
> We include the frictions between cloth and air, which is common in daily life,
> in both the generated data and modeling process.
> As shown in Table 1 and Figure 3 in the main text,
> with the modeling of dissipation by $\Phi_d$,
> our EUNet achieves lower errors
> and delivers more reasonable energy mappings as the cloth deforms,
> suggesting the effectiveness of $\Phi_d$.
>
> In addition,
> since collision forces are independent of the constitutive behaviors,
> we do not model collisions as part of our EUNet.
> We avoid the self-collisions in training data by setting an upper bound on the initial cloth's velocities and the range of their directions.
>
> Quantitative results in Table 1 of the main manuscript
> and the performance on garment animations demonstrate the effectiveness of our EUNet.
>
> # W3: More Evaluations to Verify Learned Constitutive Model
> Since the final goal is learning to animate the garments
> where EUNet is key module of this framework,
> we conduct comprehensive experiments on garment animations in Table 2 and Figure 4 in the main text.
>
> In addition,
> we further implement HOOD [1] as an extra baseline.
> Please refer to the PDF in global rebuttal for more details about the performances and the differences between baselines.
> As shown in Table 2 of both main manuscript and the global PDF,
> either MGN-S or MGN-H (used in HOOD) being the neural simulator architecture,
> simulators constrained by our EUNet always outperforms baselines (i.e., HOOD and others in the Table 2 of the global PDF),
> indicating the effectiveness of our EUNet.
>
> [1] Artur Grigorev, et al. HOOD: hierarchical graphs for generalized modelling of clothing dynamics. CVPR2023
>
> # W4: Comparisons with HOOD and Analytical Clothing Model
> Firstly,
> as discussed in W3,
> we further compare with HOOD, where simulators constrained by our EUNet outperforms it.
>
> As for the claim 'better than the analytical constitutive model',
> it is worth noting that we didn't claim it in abstract and introduction. We only mentioned this in our experimental section as an empirical analysis at L298-299. We will clarify this in the revision.
>
> Moreover,
> to further verify the advance of our EUNet over analytical models is not caused by the limited accuracy of neural simulator,
> we pair our EUNet and the analytical models with two different neural simulator architectures, namely MGN-S and MGN-H.
> As a result,
> simulators constrained by our EUNet always achieve lower errors,
> which means adopting our disentangled learning scheme with EUNet is better than using analytical models to train neural simulators.
>
> Thirdly,
> we further report the comparisons that with or without the mini-network in Table 2 from the global PDF.
> The mini-network enables the unsupervised simulators,
> which are constrained by analytical physics model,
> to obtain higher accuracy.
>
> Finally,
> we admit that implementing numerical simulations or PDE solvers within a short period is challenging. However,
> within the scope of learning to animate garments
> and based on the experiments,
> our method can better enhance learning-based simulators
> to achieve lower errors and faithful animations.

---

> > ### Comment · Reviewer_QaVe · 2024-08-11
> >
> > I appreciate the efforts made by the author in the rebuttal.
> >
> > On the formulation side, the rebuttal makes it clear that it avoids collision and friction in both the modeling and the data. This makes more sense to me now. However, I agree with Reviewer cPXz that this setting would make it too ideal for capturing the constitutive model of a **real** garment, which is where the method can be truly interesting.
> >
> > On the result side, a comparison with HOOD surely makes the validation more solid. However, it is still my opinion that using the model to constrain a neural simulator is less convincing than showing it works with a full numerical simulator, and yet better, being better than an analytical constitutive model in that case.
> >
> > Given these considerations, I would like to increase my rating to Borderline Accept. While the paper still has some limitations, I find the idea of learning a constitutive model for an unknown analytical form interesting. I wish it can become useful in the real world one day in the future.

---

> > > ### Author Response · Authors · 2024-08-11
> > >
> > > We thank the reviewer for the positive feedback and recognition of our contributions and efforts. We would like to provide additional comments on the correspondence between our work's settings and potential real-world scenarios.
> > >
> > > ## Difficulties in Collecting Real Data
> > > As demonstrated in [1],
> > > existing techniques are able to capture temporally consistent data from real world.
> > > The settings in [1] closely resemble ours and include the dynamics where a cloth is hanged in the air.
> > > Therefore, the data required by our method is feasible to capture in real-world scenarios.
> > >
> > >
> > > ## Simplification of Non-Collision Settings
> > > As shown in the visual demos from official websites of [1],
> > > with properly designed external forces,
> > > it is feasible to capture cloth dynamics without self-collisions.
> > >
> > >
> > > ## Simplification of Using Only One Piece of Cloth
> > > The simplification is reasonable, as garments and clothes made of the same materials exhibit the same constitutive behaviors. Therefore, learning the constitutive behaviors from garments is equivalent to learning from a single piece of cloth. As stated in our abstract and global rebuttal, the approach of using a piece of cloth to learn garment dynamics reduces the great need of a large scale of garment data.
> > >
> > > Besides,
> > > the settings of learning from a piece of cloth
> > > is applicable in real scenarios such as [1].
> > > Comparing with collecting data from entire garments,
> > > it is easier to handle a piece of cloth and more feasible to obtain non-collision data.
> > >
> > > At last,
> > > we agree that including collisions,
> > > such as self-collisions within clothes and collisions with other objects,
> > > would lead to more complex settings that exist in real life.
> > > However,
> > > we also want to emphasise the **reasonable simplifications** without collisions is **applicable and effective** for learning constitutive models.
> > > We would include the collisions and more complex settings in the future.
> > >
> > > [1] David Blanco Mulero, et al. Benchmarking the Sim-to-Real Gap in Cloth Manipulation. IEEE Robotics Autom2024

---

### Official Review · Reviewer_TX4H · 2024-07-11

**Soundness:** 3
**Presentation:** 2
**Contribution:** 3
**Rating:** 5
**Confidence:** 4

**Summary:**

The paper proposes a novel method for animating garments by learning from a single piece of cloth. This approach circumvents the need for large-scale garment datasets, which are resource-intensive and time-consuming to create. The core idea is to use a disentangled scheme where constitutive behaviors are learned from observed cloth and then applied to animate various garments. The proposed Energy Unit Network (EUNet) captures constitutive relations in the form of energy, bypassing the need for traditional physics models.

**Strengths:**

The paper introduces a novel disentangled approach that separates the learning of constitutive behaviors from the animation process.

 The EUNet models constitutive behaviors using energy units, allowing for direct learning from observed cloth trajectories without traditional physics models.

The approach significantly reduces the data requirement, relying on a single piece of cloth for training, making it more practical and less resource-intensive.

The method produces animations that are both robust and generalizable, capable of handling various garment types and materials.

**Weaknesses:**

The energy optimization process, although effective, can be computationally intensive and may require fine-tuning to achieve optimal results.

The paper would benefit from more extensive experimental validation, including comparisons with a broader range of existing methods and more diverse garment types.

**Questions:**

How does the performance of EUNet compare with traditional physics-based models in terms of computational efficiency and accuracy?

What are the limitations of using a single piece of cloth for training, and how can these limitations be mitigated in future work?

**Limitations:**

Comparisons with a broader range of existing methods would be appreciated, for example, one of the SOTA method named 'HOOD: hierarchical graphs for generalized modelling of clothing dynamics' which has been cited in the paper is worth comparing with.

---

> ### Author Rebuttal · Authors · 2024-08-07
>
> We thank the reviewer for the positive and valuable feedback.
>
> # W1: Computationally Intensive Energy Optimization Process
> In this paper,
> we primarily focus on the challenge of modeling constitutive relations from observations.
> The energy-based simulation is used solely as a tool to solve the dynamics constrained by our EUNet.
> Consequently,
> accelerating the energy optimization process is beyond the scope of our work.
>
> # W2: Extensive Experimental Validation
> As requested by the reviewer,
> we implement HOOD [1] for further comparisons.
> Moreover,
> the baselines we have chosen are representative:
> MGN is the pioneering approach for animating mesh-based cloth,
> LayersNet achieves state-of-the-art performance for garment animation trained in a supervised manner,
> and MGN-S+PHYS employs a recent advanced unsupervised learning scheme as shown in SNUG [2] and HOOD.
> The main difference between MGN-S+PHYS and HOOD is that HOOD adopts a hierarchical graph neural network.
> Please refer to the additional PDF in the global rebuttal for more details.
> Simulators constrained by our EUNet achieve superior performance over other baselines.
>
> In addition,
> all garments in Cloth3D [3] differ from each other in terms of length, size, and topology.
> Each category in Table 2 of the main manuscript includes several subclasses.
> For example,
> the "T-shirt" category also includes "Jackets" as shown in Figure 1 of the main manuscript,
> and the "Dress" category includes both short and long dresses.
> "Jumpsuits" and "Dresses" are combinations of upper and lower garments.
>
> [1] Artur Grigorev, et al. HOOD: hierarchical graphs for generalized modelling of clothing dynamics. CVPR2023
>
> [2] Igor Santesteban, et al. SNUG: Self-Supervised Neural Dynamic Garments. CVPR2022
>
> [3] Hugo Bertiche, et al. CLOTH3D: Clothed 3D Humans. ECCV2020
>
> # Q1: Efficiency and Accuracy Comparing with Physics-based Models
> As shown in global rebuttal,
> our EUNet is comparable to the traditional clothing model in terms of speed.
> While our EUNet contains two branch: $\Phi(\cdot)$ for potential energy and $\Phi(\cdot)$ for dissipation,
> each branch is slightly faster than the traditional clothing model.
>
> In terms of accuracy,
> as shown in Table 2 of both the main manuscriptand the global PDF, simulators constrained by our EUNet achieves higher accuracy than those constrained by physics-based clothing models.
> Furthermore,
> our method reduces the need to carefully select different types of physics-based clothing models and estimate the corresponding parameters.
> Instead,
> our EUNet directly captures the observed constitutive laws,
> enabling learning-based simulators to achieve superior performance in garment animation.
>
>
> # Q2: Limitations of Using A Piece of Cloth
> One limitation is that a single piece of cloth may not encompass all possible deformations and corresponding dynamics.
> For example,
> shear deformation may be less obvious,
> and interactions among different layers of clothing are unavailable.
> A simple solution to enrich the deformations and dynamics is to apply known forces,
> such as those controlled by robots,
> and to use multi-layered clothing during data generation to explore these interactions
>
> On the other hand,
> we intentionally design the training data to be as simple as possible
> to ensure its applicability in real scenarios.
> An example is the data from [1],
> which can be directly adopted by our model but currently unavailable.
>
> [1] David Blanco Mulero, et al. Benchmarking the Sim-to-Real Gap in Cloth Manipulation. IEEE Robotics Autom2024
>
> # Limitations: Comparison with HOOD
> Please refer to W2.

---

> > ### Comment · Reviewer_TX4H · 2024-08-08
> >
> > Thank you for your detailed response and the additional experiments you conducted. I appreciate the effort you have put into addressing my concerns.
> >
> > Regarding W1, I understand that the focus of your work is on modeling constitutive relations, and that the energy-based simulation is used primarily as a tool to achieve this goal. While the computational intensity of the energy optimization process remains a consideration, I acknowledge that accelerating this process is beyond the scope of your current research. It may be beneficial to include a brief discussion in the manuscript to clarify this distinction for the readers.
> >
> > Concerning W2, I appreciate the implementation of HOOD for further comparison and the comprehensive explanation of your chosen baselines. Your inclusion of HOOD, alongside the detailed breakdown of the garments in the Cloth3D dataset, enhances the robustness of your experimental validation. The additional PDF provided in the global rebuttal was also helpful in understanding the nuanced differences between the methods.
> >
> > In response to Q1, the clarification on the efficiency and accuracy of your EUNet compared to traditional physics-based models is noted. Your explanation of how EUNet simplifies the modeling process by capturing observed constitutive laws directly is compelling. This is indeed a strength of your approach, and the results showing higher accuracy further reinforce the value of your method.
> >
> > Finally, regarding Q2, the limitation of using a single piece of cloth is acknowledged. The potential impact on the range of deformations and dynamics captured is a valid consideration, and your proposed solution of applying known forces and using multi-layered clothing is a practical approach to addressing this. I appreciate the intentional design of your training data to ensure its real-world applicability, which is an important aspect of your work.
> >
> > Overall, the author's responses have clarified many of my concerns, and I recognize the contributions your work makes to the field. However, after careful consideration, my overall assessment and rating of the paper will remain the same.

---

> > > ### Author Response · Authors · 2024-08-09
> > > **Official Comment by Authors**
> > >
> > > We thank the reviewer for the positive feedback and recognition of our contributions and efforts.
> > >
> > > Regarding W1,
> > > existing techniques for optimizing garment dynamics with 2k to 12k vertices driven by human bodies with 7k vertices,
> > > such as HOOD,
> > > take approximately 26 hours on an NVIDIA Quadro RTX 6000.
> > > We will clarify the relationship between our contributions and the energy optimization process in the revised version.
> > >
> > > We hope the reviewer will reconsider raising the score, as we have addressed your concerns. Please let us know if there are any remaining issues.

---

### Official Review · Reviewer_Hyzo · 2024-07-18

**Soundness:** 1
**Presentation:** 2
**Contribution:** 1
**Rating:** 4
**Confidence:** 4

**Summary:**

This paper proposes to learn garment dynamics using a disentangled learning framework and the Energy Unit Network (EUNet). Instead of relying on extensive garment datasets, the approach learns constitutive behaviors from a single cloth piece and dynamically animates garments through energy optimization.

**Strengths:**

The writing is clear and technical details are described clearly.  The visual aids and diagrams are well-integrated, enhancing understanding.

**Weaknesses:**

My main problem with the paper is that the problem of learning/recovering cloth dynamics from structured sample tests has been studied intensively for a long time, and the authors seem not to be aware of this whole field.  This is a well-studied problem, and the authors do not position their method against significant prior works. Many existing works have also attempted learning from real-world fabric sample tests or indirect representations (video), which is a much harder problem. To mention a few:

1. "Predicting the Drape of Woven Cloth Using Interacting Particles" Breen et al., 1994
2. "Estimating Cloth Simulation Parameters from Video" Bhat et al., 2003
3. "Data-driven elastic models for cloth: Modeling and measurement" Wang et al., 2011
4. "How Will It Drape Like? Capturing Fabric Mechanics from Depth Images" Rodriguez-Pardo et al., 2023
5. "Estimating Cloth Simulation Parameters From Tag Information and Cusick Drape Test" Ju et al., 2024

The authors should thoroughly review the literature and reposition their contribution and provide experimental comparisons against existing works.  Additionally, the literature review section does not include important works from the physics simulation community. Including these references and discussing how the proposed method builds upon or differs from them would strengthen the paper significantly.

**Questions:**

1. Literature Positioning: Can you clarify your awareness and positioning of your method in relation to the existing body of work on learning/recovering cloth dynamics from structured sample tests? As discussed above, many significant studies in this area were not referenced.
2. Physics Simulation/Graphics Community References: The literature review section did not include important works from the physics simulation and graphics community. How does your approach relate to or differ from the significant contributions in this field? Including a discussion of this could provide a better contextual grounding for your work.
3. Experimental Validation: Can you provide more details on how your experiments validate the proposed method against these existing works? Specific comparisons and metrics would help clarify the effectiveness and novelty of your approach. Can you validate your approach against different datasets, including synthetic datasets generated from different simulation engines as well as real-world datasets used by current works?

**Limitations:**

The authors provided a discussion on the limitations of edge-wise discretization and a lack of self-collision handling.

---

> ### Author Rebuttal · Authors · 2024-08-07
>
> We thank the reviewer for the detailed feedback.
>
> # W1: Insufficient Literature Review
> We argue that our focus is quite different from **Physics Parameter Estimation**.
> The misunderstanding may come from the similarity in data format,
> where a piece of hanged cloth deforms given external forces.
> Unlike physics parameter estimation,
> our EUNet aims at learning the unknown **ENERGY FUNCTION** that describes the constitutive laws. While in physics parameter estimation, the constitutive models are **known a priori and carefully chosen**.
>
> Specifically,
> with known constitutive models, physics parameter estimation aims at learning physics parameters, such as the lame constant in St. Venant-Kirchhoff elastic model for the target material.
> Complex devices and procedures, such as KES-F [1], the FAST system [2], and the Cusick drape test [3] are commonly required to estimate the parameters that best fit the analytical cloth models or simulators.
> Moreover,
> for a specific material, such as cotton,
> different clothing models or simulators require separate procedures or pipelines to estimate different sets of physics parameters.
>
> In contrast,
> our EUNet directly captures constitutive behaviors of different materials, which takes deformation attributes (i.e., delta of length and bending theta) and material attributes (i.e., stiffness and damping coefficients) as input,
> and outputs the corresponding potential energy.
> The material attributes remain constant in our study.
> As discussed at L84-88,
> our EUNet does not need existing analytical physical models and physics parameter estimation.
>
> In terms of literature review,
> as mentioned above we focus on learning to animate garments where modeling constitutive behaviors is the key challenge,
> we thus mainly discuss the garment animation and constitutive laws in the literature review section at L108-126 and L127-136 respectively.
> Energy-based physics simulation serves as a tool to generate dynamics constrained by our EUNet,
> and we briefly introduce the corresponding techniques at L137-144.
>
> We will clarify the relationship between our focus and physics parameter estimation in the revision.
>
> [1]. Kawabatak, et al. Fabric performance in clothing and clothing manufacture. Journal of the Textile Institute. 1989
>
> [2]. Minazio, et al. FAST–fabric assurance by simple testing. IJCST1995
>
> [3]. Cusick, et al. The dependence of fabric drape on bending and shear stiffness. Journal of the Textile Institute Transactions 1965.
>
> # Q1: Literature Positioning
> Please refer to W1.
>
> # Q2: Physics Simulation/Graphics Community References
> As discussed in W1,
> our focus is on how to learn the constitutive laws,
> which is discussed at L127-136 and involves several references from graphics community.
> We verify our EUNet by applying existing physics simulation techniques,
> which we briefly discussed at L137-144.
>
> # Q3: Experimental Validation
> As discussed in W1,
> our work is orthogonal to physics parameter estimation,
> thus comparing with methods mentioned by the reviewer is unnecessary.
>
> In addition,
> as discussed in W1 and Q2,
> our goal is to faithfully animate garments with observed materials where modeling constitutive relations is the key challenge.
> We thus verify the effectiveness of our EUNet by integrating with simulation techniques and evaluate the garments animations on Cloth3D [1], which is a large scale dataset and commonly used in garment animations, such as DeePSD [2] and MotionGuided [3].
>
> [1] Hugo Bertiche, et al. CLOTH3D: Clothed 3D Humans. ECCV2020
>
> [2] Hugo Bertiche, et al. DeePSD: Automatic deep skinning and pose space deformation for 3D garment animation. ICCV2021
>
> [3] Meng Zhang, et al.  Motion guided deep dynamic 3D garments. TOG2022

---

> > ### Comment · Reviewer_Hyzo · 2024-08-13
> >
> > Thank you for the detailed rebuttal and for clarifying the focus and contributions of your work, particularly the distinction between your approach and traditional physics parameter estimation.  Based on your clarifications, I will be raising my score.
> >
> > Given that your method aims to learn constitutive models, I believe it would be beneficial to include demonstrations on real data to further substantiate the practical applicability of your approach. This addition could strengthen the impact of your work.

---

> > > ### Author Response · Authors · 2024-08-13
> > >
> > > We thank the reviewer for the reply.
> > > Our work aims at learning to animate garments,
> > > where modeling the underlying constitutive behaviors is the core challenge.
> > >
> > > We actually already provide demonstration on real data.
> > > Specifically,
> > > although real data for learning constitutive laws are not publicly available yet,
> > > simulators constrained by our EUNet,
> > > which learns the constitutive behaviors on simulation data,
> > > **can directly be used** to animate real garments in CAPE [1],
> > > as shown in Figure 2 of the global rebuttal.
> > >
> > > In addition,
> > > it is **standard practice** to benchmark different methods on simulation data in both physics simulation [2, 3] and cloth(garment) animation [4, 5, 6, 7].
> > >
> > > [1] Qianli Ma, et al. Learning to Dress 3D People in Generative Clothing. CVPR2020
> > >
> > > [2] Tobias Pfaff, et al. Learning mesh-based simulation with graph networks. ICLR2021 (**Outstanding Paper**)
> > >
> > > [3] Yitong Deng, et al. Fluid Simulation on Neural Flow Maps. TOG2023 (**Best Paper**)
> > >
> > > [4] Chaitanya Patel, et al. TailorNet: Predicting Clothing in 3D as a Function of Human Pose, Shape and Garment Style. CVPR2020
> > >
> > > [5] Hugo Bertiche, et al. CLOTH3D: Clothed 3D Humans. ECCV2020
> > >
> > > [6] Meng Zhang, et al. Motion guided deep dynamic 3D garments. TOG2022
> > >
> > > [7] Xingxing Zou, et al. Cloth4d: A dataset for clothed human reconstruction. CVPR2023

---

### Author Rebuttal · Authors · 2024-08-07

We thank the reviewers for their insightful feedback.
We emphasize our contributions and clarify the main points as follows.


To mimic the dynamic patterns from observed clothes, some methods [1, 2] focus on estimating the **PHYSICS PARAMETERS** that best fit the known analytical models or simulators.
In contrast,
we aim to explore the intrinsic **ENERGY FUNCTIONS** that describe the constitutive relations governed by general physics laws, such as the Lagrangian mechanics,
and animate garments constrained by our learned EUNet.

Our work is significant in the following ways
1. our disentangled learning scheme reduces the great need of large scale of garment data to mimic garment dynamics, and relies on only a piece of cloth as training data;
2. our EUNet is able to directly capture the constitutive behaviors from observed trajectories that are relatively easy to obtain in real life,
and is highly generalizable thanks to the topology-independent designs;
3. garment animations constrained by our EUNet deliver lower errors and superior performance comparing with baselines.

In the additional PDF file,
we
1. add HOOD [3] for comparisons and demonstrate the effectiveness of the mini-network mentioned at L290-292 for HOOD;
2. apply our method in real application and animate the realistic garments from CAPE [4].

Below we list the key components of the tables.

# Time Efficiency Comparisons
|        | PHYS  | EUNet $\Phi_p$ | EUNet $\Phi_d$ | EUNet $\Phi_p+\Phi_d$ |
|  ----  | ----  | ----  | ----  | ----  |
|  Time (ms)  | 1.743 $\pm$ 0.212  | 1.055 $\pm$ 0.202 | 1.216 $\pm$ 0.223 | 2.271 $\pm$ 0.422 |

We denote the StVK elastic model and the bending model by "PHYS", which is used in the main text.
As formulated in Equation (1),
our EUNet is composed of two separate branches: $\Phi_p$ for potential energy and $\Phi_d$ for dissipation.
Both branches have the same structures.
We report the time separately for each branch and the full EUNet as above.
The forward time is averaged on 80 frames of predictions,
which include garments composed of 7924 vertices, 23636 edges and 15712 faces.
All experiments are run on NVIDIA A100-SXM4-80GB.

Our EUNet is comparable to the traditional clothing model in terms of speed.

# Comparisons with HOOD
|     Methods   | Overall Euclidean Error (mm)  | Collision (%) |
|  ----  | ----  | ----  |
|  MGN-H+PHYS+F (HOOD) | 84.85 $\pm$ 29.93  | 0.46 $\pm$ 0.99 |
|  MGN-H+EUNet (Ours) | **66.39 $\pm$ 39.36**  | **0.44 $\pm$ 0.48** |

We denote the hierarchical graph neural network used in HOOD as MGN-H.
HOOD adopts the analytical clothing model (PHYS) and friction (F) between human body and garments as loss terms.
As shown in the table,
MGN-H constrained by our EUNet achieves lower errors comparing with HOOD,
suggesting the effectiveness of our EUNet.

[1] Carlos Rodríguez-Pardo, et al. How Will It Drape Like? Capturing Fabric Mechanics from Depth Images. CGF2023.

[2] Eunjung Ju, et al. Estimating Cloth Simulation Parameters From Tag Information and Cusick Drape Test. EG2024.

[3] Artur Grigorev, et al. HOOD: hierarchical graphs for generalized modelling of clothing dynamics. CVPR2023

[4] Qianli Ma, et al. Learning to Dress 3D People in Generative Clothing. CVPR2020

---

### Decision · Program_Chairs · 2024-09-25

**Decision:**

Accept (poster)

**Comment:**

The paper received mixed scores (WA, BA, BA, BR). During the rebuttal phase, the authors and reviewers discussed the paper. Unfortunately, the final scores remained mixed and lukewarm at best, without any reviewer championing the work. The AC reviewed the paper, reviews, and discussion and decided to champion the work.

+ significantly reduces the requirement for generating/collecting data over all possible garment/body configurations
+ a nice formulation involving energy formulation (this is different from parameter estimation) giving a fresh perspective to the problem
+ a novel direction and can lead to followups (this is related to gradient-domain processing in imaging and animation)

- lack of demonstration on real data (future work!)
- not demonstrated on very complex samples or real captures. AC finds this to be ok, given the novelty of the approach.